# Hulled Wheat Productivity and Quality in Modern Agriculture Against Conventional Wheat Species

**Leszek Rachoń [1], Aneta Bobryk-Mamczarz [2] and Anna Kiełtyka-Dadasiewicz [1,*]**

1    Department of Plant Production Technology and Commodity Science, University of Life Sciences in Lublin, Akademicka 15, 20-950 Lublin, Poland; leszek.rachon@up.lublin.pl
2    PZZ LUBELLA GMW Sp. z o.o. Sp. k., ul. Wrotkowska 1, 20-469 Lublin, Poland; A.Bobryk-Mamczarz@maspex.com
*    Correspondence: anna.kieltyka-dadasiewicz@up.lublin.pl; Tel.: +4881-445-6629

**Abstract:** The objective of this study is to compare the yields and qualities of the hulled wheats emmer (*Triticum dicoccum* Schübl.) and spelt (*Triticum aestivum* L. ssp. *spelta*) with the commonly cultivated naked wheats common wheat (*Triticum aestivum* L. ssp. *vulgare*) and durum wheat (*Triticum durum* Desf.). Three years of field experiments were carried out from 2015 to 2017 in the Lubelskie province (Poland) on rendzina soils. The experimental results indicate that the hulled wheats, even when cultivated with advanced technology, produced lower yields compared to the common and durum wheats (reduced by 30–56%). In spite of their lower yields, emmer and spelt retained appropriate technological parameters. Higher ash, protein, and wet gluten yields were characteristic of the hulled wheats; however, the high gluten spread of emmer (13.3 mm) may limit its application as a raw material in some food processes. In summary, hulled wheat species can be recommended for modern agricultural production as an alternative source of high-quality materials for the agricultural and food industries.

**Keywords:** common wheat; durum; spelt; emmer; yield; quality

---

## 1. Introduction

Cereals are among the oldest components of the human diet. They comprise a group of crop plants that occupy approximately one-half of the total agriculturally cultivated area globally. According to data from the Food and Agriculture Organization of the United Nations, the average annual production of cereals worldwide is about 2.5 billion tons, including approximately 750 million tons of wheat. Wheat accounts for about 30% of the total cultivation area of cereals in the world, corresponding to approximately 220 million hectares. About 60% of the total production of wheat is used for consumption purposes [1].

The globalization of production and a growing demand for products with high health-promoting values has induced the food industry to search for new technological solutions. One of these is a return to "ancient", high-quality wheat species which can enhance biodiversity and the nutritive values of products [2,3].

In spite of their lower yields, hulled wheat species such as spelt, einkorn, or emmer are characterized by high protein content and gluten yield, and notably exceed common wheat and even durum wheat in terms of their contents of macro and micro-elements and vitamins [4–6]. Additional advantages of these wheat species are their lower habitat requirements and higher tolerance to disease [7]. Emmer wheat (*Triticum dicoccum* Schübl.) is a primitive and truly ancient species of tetraploid wheat, which, in ancient times, was considered as the most valuable cereals. It was one of the earliest cultivated wheat species (just after einkorn), is rich in proteins and minerals, and is valued for its excellent taste [8–10].

Spelt wheat (*Triticum aestivum* ssp. *spelta*) is a hexaploid, hulled, and non-threshable wheat species with high nutritive, dietetic, and taste value. Spelt was enormously important in Europe as early as the Bronze Era, and later in the Middle Ages [11].

Suchowilska et al. [12] observed that the contents of minerals (especially microelements), vitamins, and organic compounds necessary for correct development of the organism decrease in wheat along with increases in the productivity and yields of its genotypes, which can lead to highly unfavourable consequences. At present, sustainable agriculture is preferred over yields, and hulled wheat species can provide an excellent alternative for producers of high-quality foods.

The results of studies comparing the contents of nutrients, qualities, and suitability of various wheat species in the food industry are commonly known. However, as a rule, the materials used in these studies originated from cultivation conducted in varying soil and climate conditions. Hulled wheat species (i.e., emmer and spelt) are usually dedicated as primary crops in organic or small-area farms and are generally tolerant of less-advanced cultivation technologies. On the other hand, common wheat and durum wheat are typical industrial raw materials, and hence, are usually cultivated in large production farms. For this reason, we do not really know whether the results of qualitative studies are only affected by species and variety differences, or are also affected by climate, soil, and cultivation technology differences [3].

The objective of this study is to compare the yields and qualities of hulled wheat species (i.e., emmer wheat and spelt wheat) with commonly grown naked wheat species (i.e., common wheat and durum wheat). The presented results were obtained by conducting a strict field experiment with identical cultivation conditions for all the above-mentioned wheat species, while the applied cultivation technology conformed with the principles of cereal production in contemporary agriculture.

## 2. Materials and Methods

### 2.1. Experiment Location and Conditions

The field experiment was conducted from 2015 to 2017 in the locality of Hopkie (50°30′28′′ N 23°39′40′′ E; 221 m a.s.l., Lubelskie Province, Poland) on rendzina soil formed from the transformation of calcareous rock with high levels of silts (~68%), sand (~22%), and clay (~10%). In the agronomic classification, the category of heavy mineral soils has been identified as being from the good wheat complex (soil quality class II). The soil has an alkaline reaction ($pH_{KCl}$ = 7.4) and an analysis of the available components in the range of macro-elements revealed very high contents of phosphorus (P = 0.23 g kg$^{-1}$) and potassium (K = 0.26 g kg$^{-1}$), and a medium content of magnesium (Mg = 0.051 g kg$^{-1}$). The experiment was set up using a random block design, with three replicates, on plots with area of 0.17 ha. The experimental material consisted of four spring wheat (*Triticum* L.) cultivars:

- *T. aestivum* L. ssp. *vulgare*-cv. "Toridon."
- *T. durum* Desf.-cv. '"loradur."
- *T. aestivum* L. ssp. *spelta*-cv. "Wirtas."
- *T. dicoccum* Schübl.-cv. "Bondka."

Each year, the conducted soil tillage was typical of a conventional tillage system. The fore crop was sugar beet. After its harvest, pre-winter ploughing was performed. In spring, the first treatment was harrowing, with an application of phosphorus–potassium fertilization at the doses of 17 kg·ha$^{-1}$ and 50 kg·ha$^{-1}$, respectively. Next, under conditions of optimum soil moisture, finishing tillage was applied using a tillage set. Prior to sowing, seeds were primed with the preparation Kinto Duo 080 FS (tritikonazol + prochloraz complex with copper) in an amount of 200 mL per 100 kg of sowing grain. The sowing, at the rate of 450 kernels per 1 m$^2$ (180–220 kg, depending on the weight of 1000 kernels), took place at the optimum agrotechnical time, from 20 March to 4 April.

Protection treatments against diseases and pests were performed uniformly for all wheat species and in all consecutive years, in conformance with the recommendations for production plantations.

The plant protection treatments consisted in chemical control of monocotyledonous and dicotyledonous weeds. In the full tillering phase (BBCH-21-23), a mixture of the herbicides Lintur (Syngenta Crop Protection Sp. z o.o., Warszawa, Poland), 70WG (dicamba + triasulfuron), and Chwastox Extra 300 SL (MCPA CIECH Sarzyna S.A., Nowa Sarzyna, Poland) was applied, in amounts of 0.15 kg ha$^{-1}$ and 0.95 L·ha$^{-1}$, respectively. At the beginning of the shoot development phase (BBCH-31-33), Axial 50 EC (pinoxaden) was applied in an amount of 0.6 L ha$^{-1}$. To provide protection against lodging, in the phase of the first node (BBCH-31), Moddus 250 EC (trinexapac-ethyl) and Stabilan 750 SL (chloromequat chloride) were applied in the amounts of 0.26 and 0.57 L ha$^{-1}$, respectively. For protection against fungal diseases, plants were treated with the fungicide Duett Starr 334 SE (fenpropimorph + epoxiconazole) in the amount of 1 L ha$^{-1}$ during the phase of shoot development (BBCH-29-31) and with the preparation Prosaro 250 EC (prothioconazole + tebuconazole) in the phase of heading (BBCH-51-59), in the amount of 0.99 L ha$^{-1}$. In the period of pest occurrence, the insecticide Fury 100 EW (zeta cypermethrin) was applied in the amount of 0.08 L ha$^{-1}$. Nitrogen fertilisation, 150 N kg·ha$^{-1}$ in total, was applied three times: before sowing (80 kg ha$^{-1}$), in the phase of shoot development (35 kg ha$^{-1}$), and in the phase of heading (35 kg ha$^{-1}$).

## 2.2. Yield and Biometric Determination

Harvest was performed with the use of a Claas Mega 208 combine harvester (Claas, Harsewinkel Germany) in the first ten days of August, in the phase of full ripeness. After the harvest, the grain was cleaned, dried, and determinations of grain yield and biometric features were carried out.

The following parameters were estimated:

- Grain yield (t·ha$^{-1}$);
- Weight of 1000 kernels WTK (g), counting 2 × 500 kernels, according to PN-R-74017:1968;
- Grain test weight (kg·hL$^{-1}$), according to the standard PN-73/R-74007;
- Grain vitreousness (%), according to the standard PN-70/R-74008, with the use of a farinotom;
- Grain colour (parameter b *), assessed with the colorimetric method using a Konica–Minolta Chroma Meter CR-410, in the L * a * b * system done on whole grain samples.

## 2.3. Qualitative Analysis

The grain of all analysed wheat species was estimated, in terms of:

- Total protein content (%): total nitrogen content was determined with the Kjeldahl method and converted to protein (using the factor of 5.70), according to the standard PN-EN ISO 20483:2007;
- Yield of wet gluten (%), according to the standard PN-77/A-74041;
- Gluten spread (mm), according to the standard PN-77/A-74041;
- Total ash content (%), according to the standard PN-ISO 2171.

## 2.4. Statistical Analysis

The results were processed statistically with the method of analysis of variance (ANOVA) using the Statistica 12 PL (StatSoft Polska: Kraków, Poland) software. Differences were estimated using Tukey's post-hoc HSD (honest significant difference) test at a significance level of $p \leq 0.05$. To determine the correlations between the analysed traits, Pearson's linear correlation analysis was performed. Furthermore, the coefficients of variation were calculated for the wheat cultivars as follows:

$$CV (\%) = (Std/x) \times 100 \qquad (1)$$

where Std is the standard deviation and x is the mean. Presentation of the results and their statistical analyses were performed using Excel spreadsheets.

## 2.5. Agro-Meteorological Conditions

Agro-meteorological conditions in the growing season were determined as average monthly temperatures (Table 1) and total precipitation (Table 2), based on data from meteorological stations.

**Table 1.** Mean monthly air temperatures (°C) in the vegetation seasons of 2015–2017, according to data from the meteorological station at the District Authority in Tomaszów Lubelski against mean values of air temperatures in the years 1951–2010, as measured at the meteorological station in Felin, Lublin (University of Life Sciences).

| Months of Vegetation | Years | | | |
|---|---|---|---|---|
| | 2015 | 2016 | 2017 | 1951–2010 |
| March | 8.2 | 3.9 | 5.9 | 1.0 |
| April | 8.3 | 9.7 | 8.3 | 7.4 |
| May | 13.3 | 14.9 | 14.3 | 13.0 |
| June | 18.1 | 13.4 | 19.0 | 16.3 |
| July | 20.3 | 20.3 | 19.2 | 18.0 |
| August | 21.8 | 18.7 | 20.2 | 17.2 |
| Mean temperature | 15.0 | 13.5 | 14.5 | 12.2 |

**Table 2.** Distribution of precipitation (mm) in vegetation seasons of 2015–2017, according to measurements at RSP (Rolnicza Spółdzielnia Produkcyjna) Hopkie against measurements of precipitation totals in individual months in the period of 1951–2010 at the meteorological station in Felin, Lublin (University of Life Sciences).

| Months of Vegetation | Years | | | |
|---|---|---|---|---|
| | 2015 | 2016 | 2017 | 1951–2010 |
| March | 11.0 | 47.5 | 36.5 | 28.0 |
| April | 45.0 | 79.5 | 31.0 | 39.0 |
| May | 48.0 | 69.0 | 66.0 | 60.7 |
| June | 15.0 | 66.0 | 41.0 | 65.9 |
| July | 89.5 | 63.5 | 54.5 | 82.0 |
| August | 5.5 | 40.0 | 32.5 | 70.7 |
| Precipitation total | 214.0 | 365.5 | 261.5 | 346.3 |

For the estimation of the hydrothermal conditions and for more accurate estimation of the effect of atmospheric conditions on the growth and development of wheat in the analysed vegetation seasons, Selyaninov's hydrothermal coefficient ($H_c$) was calculated, using the following formula:

$$H_c = \frac{h_n}{0,1n \times \overline{x}_t}, \tag{2}$$

where $h_n$ is the monthly total of atmospheric precipitation (in mm), $n$ is the number of days in a given period, and $\overline{x}_t$ is the sum of mean diurnal air temperatures (in °C).

The results obtained correspond with the weather conditions, as per Tables 3 and 4; correlation strength follows the increasing intensity of blue. Calculation of the values of Selyaninov's hydrothermal coefficient for the individual vegetation seasons produced the following:

**Table 3.** Values of Selyaninov's hydrothermal coefficient, characterising the weather conditions in individual years and months of the experiment.

|  | $H_c$ in 2015 | $H_c$ in 2016 | $H_c$ in 2017 |
|---|---|---|---|
| March | 0.4 | 3.9 | 2.0 |
| April | 1.8 | 2.7 | 1.2 |
| May | 1.2 | 1.5 | 1.5 |
| June | 0.3 | 1.6 | 0.7 |
| July | 1.4 | 1.0 | 0.9 |
| August | 0.1 | 0.7 | 0.5 |
| Mean (season) | 0.8 (d) | 1.5 (o) | 1.0 (d) |

$H_c$: hydrothermal coefficient.

**Table 4.** Range of values of Selyaninov's coefficient and corresponding weather conditions.

| Extremely Dry (ed) | $H_c \leq 0.4$ |
|---|---|
| Very dry (vd) | $0.4 < H_c \leq 0.7$ |
| Dry (d) | $0.7 < H_c \leq 1.0$ |
| Fairly dry (fd) | $1.0 < H_c \leq 1.3$ |
| Optimal (o) | $1.3 < H_c \leq 1.6$ |
| Fairly wet (fw) | $1.6 < H_c \leq 2.0$ |
| Wet (w) | $2.0 < H_c \leq 2.5$ |
| Very wet (vw) | $2.5 < H_c \leq 3.0$ |
| Extremely wet (ew) | $H_c > 3.0$ |

Analysis of the above values of the $H_c$ coefficient confirms that a majority of months were dry in 2015, apart from the optimal July and a fairly wet April, and 2017 was similar, except for optimal conditions in May and a fairly wet March. Both of these seasons were, therefore, classified as dry ones. The year 2016 was different, as Selyaninov's hydrothermal coefficient qualified the beginning of the vegetation season as very or even extremely wet (March), the middle of the season as optimal, and the end as dry and very dry (harvest), classifying the entire vegetation season as one with optimal weather conditions. According to Dzieżyc [13], the rainfall requirements of spring wheats in the Lublin region, for medium soils, amount to 44 mm of precipitation in April, 65 mm in May, 97 mm in June, and 94 mm in July (a total of 300 mm over the vegetation season). In the case of heavy soils, the water requirements are lowered by about 10%. In this context, one can note a deficit of precipitation in June and July in all analysed years, compared to the multi-year period values, and an excessively dry April in 2017 and an excessively dry May in 2015.

## 3. Results and Discussion

### 3.1. Yields

Regardless of the year of the experiment, the highest yield was produced by common wheat (7.69 t ha$^{-1}$), with durum wheat producing a yield 17.6% lower (see Table 5). The hulled wheat species were characterised by distinctly lower yields–spelt at a level of 4.47 t ha$^{-1}$ and emmer wheat at 3.42 t ha$^{-1}$ (lower by 41.9% and 55.5%, respectively, relative to common wheat). However, for hulled wheats, the yield was stable regardless of the weather conditions. Both the emmer and spelt yields did not differ statistically in the particular years of the study, while the yields of the high-yielding wheats (i.e., common and durum) changed significantly, depending on the agro-meteorological conditions.

**Table 5.** Grain yield (t ha$^{-1}$) of four species of wheat in the years 2015–2017.

| Wheat Species | Year | | | Mean |
|---|---|---|---|---|
| | 2015 | 2016 | 2017 | |
| common | 7.67 ab# | 7.34 b | 8.07 a | 7.69 A# |
| durum | 6.01 de | 6.65 c | 6.37 cd | 6.34 B |
| spelt | 4.77 fg | 4.38 g | 4.26 g | 4.47 C |
| emmer | 3.23 h | 3.61 h | 3.42 h | 3.42 D |
| Mean | 5.37 B# | 5.68 A | 5.54 AB | - |

# Values denoted with the same lowercase letter do not differ statistically significantly ($p \leq 0.05$) between years and wheat species, while uppercase letters correspond to the same between years (in row) or wheat species (in column).

Notably lower yields of hulled wheat species have been confirmed by a majority of authors (see, e.g., Evans et al. [14], Konvalina et al. [15], Longin et al. [7], Marino et al. [16], and Rachoń et al. [17]). In the case of spelt, as in our study, other researchers have also observed lower yields relative to common wheat: Lacko-Bartošova and Otepka [18] observed 7.8–22.8% lower yields and Rachoń et al. [17] observed 36.5% lower yields. In turn, a two-year study conducted in organic farms by Cyrkler-Degulis et al. [8,19] demonstrated that average yields of the hulled wheat species spelt and emmer could attain yields at the level of common wheat. Cyrkler-Degulis et al. [8,19] conducted a study in the conditions of organic farms and noted that the yields of the two best-yielding cultivars of emmer wheat were 4.06 t·ha$^{-1}$ and 4.09 t ha$^{-1}$, which constituted 82–83% of the average yield of currently cultivated common wheat cultivars. In a study of the spelt wheat cultivar Bauländer Spelz conducted in the region of Nitra (Slovakia), Lacko-Bartošova and Otepka [18] obtained yields at a level of 6.06 t ha$^{-1}$; that is, lower than the yield of common wheat by as little as 7.8%. On the other hand, Rachoń et al. [17] obtained yields in the cultivar Schwabenkorn (in Polish conditions) at a level of 3.87 t ha$^{-1}$, constituting only 53.4% of the yield of common wheat cv. Tonacja. Australian researchers, Evans et al. [14], also conducted a study on the yields of spelt wheat and observed that, even in greenhouse conditions with optimum irrigation, the yield of spelt wheat was at an average level of 77% of the yield of common wheat in cv. Wedgetail, and in field conditions, at the level of 60–65%; in the case of cv. Kamarah, the values were lower than 55% of the yield of common wheat. The lower yielding potential of spelt wheat in relation to common wheat results, for example, from the smaller number and weight of kernels in an ear. In a study on the yield of emmer wheat, Stallknecht [20] compared the results of a three-year experiment conducted in Montana, in which emmer yields varied from 48 to 84% of the yields of common wheat. The yields of hulled wheat species have also been studied by Troccoli and Codianni [21] in south Italy (Apulia region). They obtained yields at the level of 3.54 t·ha$^{-1}$ in the case of emmer wheat and 2.80 t·ha$^{-1}$ in the case of spelt wheat. Furthermore, Marino et al. [16], in a two-year study conducted in central Italy, obtained yields of 1.51–2.37 t ha$^{-1}$ for cultivation without any nitrogen fertilisation and 3.25–3.46 t·ha$^{-1}$ for emmer wheat fertilised with 90 kg N ha$^{-1}$, thereby corresponding with the literature reports of Konvalina et al. [15] and Pagnotta et al. [22] demonstrating yields in the range of 1.8–3.4 t ha$^{-1}$.

Significant differences were noted in grain yield in the years considered in the experiment. The highest yield, regardless of species, was noted in 2016 (higher precipitation total, with uniform distribution), and the lowest was noted in 2015–a year with the lowest precipitation total over the vegetation period in the three years under analysis. In 2015, insufficient moisture and sunny weather after the March sowing limited the emergence rate, while the lower (compared to the multi-year average) precipitation totals in spring and early summer did not satisfy the water demand of the plants (shoot development phase, extremely dry June), which, in consequence, resulted in the lowest yields (lower number of kernels in ear and lowest weight of 1000 kernels). In 2016, the wetter spring had a beneficial effect on germination, emergence, and tillering, while mean air temperatures were also higher than the long-term average, which resulted in the highest grain yield. The spring of 2017 was characterised by temperature and precipitation totals close to those of the multi-year average, thereby

leading to optimal conditions conducive to the emergence and tillering of plants, thanks to which the number of kernels per ear and the weight of 1000 kernels were the highest (relative to the preceding years) and the yield was comparable to that of 2016. The summer months were warmer, with a lower precipitation total relative to the multi-year average, which was beneficial for plant ripening.

Analysing the interactions of the wheat species during the study years, one can note significant differences. The hulled wheat species (spelt and emmer) did not display significant differences in yield in the individual years, which indicates their lower dependence on the weather conditions–possibly due to their genetically lower habitat requirements. In the case of common wheat, the yield differed significantly between the years 2016 and 2017. In the case of the durum wheat, the highest yield was noted in 2016 and the lowest (which was statistically significantly different) in 2015.

### 3.2. Weight of 1000 Kernels

The weight of 1000 kernels is one of the main technological parameters indicating grain quality. In our study, the highest value of this parameter was obtained for the durum wheat (40.7 g), while common wheat had a lower value (34.4 g), as shown in Table 6. These findings are in agreement with the study by Rachoń et al. [17], in which the highest WTK was also obtained for durum wheat (50.2 g), followed by common wheat (45.8 g) and spelt wheat (44.7 g). Packa et al. [23] obtained the result of 37.6 g for the control sample of common wheat while, in the case of spelt, the weight of 1000 kernels was higher (at 44.1 g) and that of emmer wheat was lower (at 35.4 g). A similar relationship between common wheat and emmer wheat was observed by Borusiewicz and Załuski [23,24]. In the experiment presented here, the hulled wheat species had the lowest values of this parameter: 33.9 g for emmer wheat and 33.1 g for spelt wheat. Marino et al. [16], in an experiment conducted in Italy, obtained the values of 41.8 g and 56.6 g in the case of non-fertilised emmer wheat for two respective years; for fertilised emmer wheat, the corresponding values were 40.7 g and 51.8 g. Desheva et al. [25], in a study on 38 collection objects of emmer wheat, found that the WTK values varied within a very broad range: from 21.8 g to 50.0 g with a mean value of 36.0 g.

**Table 6.** Biometric traits of analysed wheat species (mean values for the years 2015–2017).

| Wheat Species | Weight of 1000 Kernels (g) | Test Weight (kg·hL$^{-3}$) | Vitreousness (%) | Colour, Parameter b * |
|---|---|---|---|---|
| Common | 34.4 [B#] | 76.1 [B] | 83.2 [C] | 13.2 [D] |
| Durum | 40.7 [A] | 82.8 [A] | 86.1 [B] | 23.0 [A] |
| Spelt | 33.1 [C] | 76.5 [B] | 79.0 [D] | 14.9 [C] |
| Emmer | 33.9 [BC] | 75.2 [C] | 91.9 [A] | 16.3 [B] |

[#] Values denoted with the same letter in columns do not differ statistically significantly ($p \leq 0.05$); b * system done on whole grain samples.

### 3.3. Grain Test Weight

According to Dexter and Myrchalo [26], the test weight of grains is one of the parameters which indicates the milling potential of grain. In our experiment, the highest value of this indicator was obtained in the case of durum wheat (82.8 kg·hL$^{-1}$). The grain test weight of spelt wheat was 76.5 kg hL$^{-1}$, only slightly higher than the test weight of common wheat. The lowest value was obtained in the case of emmer wheat (75.2 kg hL$^{-1}$; see Table 6). The test weights of 20 samples of emmer wheat grain analysed by Pagnotta et al. [22], harvested at two locations in Italy, presented a two-season mean value of 71.8 kg hL$^{-1}$. The test weight is a highly stable quality parameter of wheat grain. This is confirmed also by the calculated coefficient of variation for this parameter, which was the lowest (1.09%) in the case of common wheat, and attained the highest value of 2.60% in the case of spelt wheat (Table 7). The values of grain test weight obtained in this experiment attained the assumed

minimum for all genotypes, with durum wheat presenting the most suitable values for use as material for pasta production.

**Table 7.** Coefficients of variation (CV %) of analysed traits of wheat grain (mean values for the years 2015–2017).

| Wheat Species | Grain Yield | Weight of 1000 Kernels | Test Weight | Vitreousness | Colour (Parameter b *) | Total Protein Content | Wet Gluten Yield | Gluten Spread | Total Ash Content |
|---|---|---|---|---|---|---|---|---|---|
| Common | 5.32 | 5.63 | 1.09 | 8.41 | 4.72 | 2.58 | 9.03 | 20.27 | 3.75 |
| Durum | 4.81 | 4.17 | 1.39 | 5.27 | 8.09 | 4.13 | 3.98 | 9.99 | 7.10 |
| Spelt | 6.18 | 8.47 | 2.60 | 26.34 | 3.41 | 5.14 | 6.07 | 15.15 | 4.55 |
| Emmer | 6.16 | 2.44 | 1.74 | 1.67 | 5.26 | 2.46 | 2.98 | 8.39 | 15.08 |

### 3.4. Grain Vitreousness

According to Fu et al. [27], high grain vitreousness is a parameter with a beneficial effect on the quality (firmness) of pasta and other food products. In our study, the highest vitreousness was attained by emmer wheat (91.9%; Table 6), while lower values of this parameter were noted for durum wheat, followed by the common wheat. Spelt wheat was the least vitreous (79%). Subira et al. [28] demonstrated that durum wheat species, both old and contemporary, are characterised by vitrousness above 80%, with older cultivars having higher values of vitreousness (in the range of 90–92%). Giacintucci et al. [29] indicated that the spring form of emmer wheat is characterised by higher vitreousness than the winter form. Desheva et al. [25], in a study on 39 collection objects of emmer wheat conducted in Bulgaria, obtained grain vitreousness results in the range of 66–99%.

### 3.5. Grain Colour

According to numerous authors, grain colour is a genetic factor which is also related to climate conditions [30,31]. Apart from the analysis of the amount of yellow pigment, it can be estimated rapidly with the use of a colorimeter in the CIELab (Comission Internationale de l'Eclairage) system (colour space). The parameter b * indicates the contribution of the yellow colour, such that the higher the value of this parameter is, the more intense the yellow colouring of the sample is [27]. Among the wheat species under analysis, the highest value of parameter b * was characteristic of the durum wheat grain–it was 23.0 compared to 13.2 to 16.3 for other analysed wheat species ((23), Table 6). A significantly lower value of this parameter was noted for the emmer wheat grain, which was 30% lower than that of the durum wheat grain. A value even lower was noted in the case of spelt wheat grain (14.9). The lowest value (i.e., the least yellow colour) was noted for the common wheat grain–the value of the parameter b * was at the level of 13.2, 43.1% lower than the value obtained for durum wheat.

Fu et al. [27], in their study on semolina from CWAD (Canadian Western Amber durum) wheat species, obtained values of the parameter b * in the range of 27.8–32.7; Sieber et al. [32], in a study on 46 lines of durum wheat harvested in Germany, obtained b * values in the range of 15.0–19.1. Subira et al. [28], in whole-grain flours from durum wheat cultivated in the conditions of Italy and Spain, obtained b * values in the range of 12.9–14.5, the values being higher in contemporary species compared to older ones. Piergiovanni et al. [33] compared the carotenoid contents in three wheat species and found the highest average level in durum wheat, followed by spelt wheat, with the lowest in emmer wheat. Fuad and Prabshankar [34] performed colorimetric analysis of samples of semolina from various wheat species and obtained the highest value of parameter b * for durum semolina (at 20.1), followed by common wheat semolina (14.9); the lowest value was found in emmer wheat (14).

### 3.6. Protein Content and Gluten Yield and Quality

Protein content in wheat grain is one of the main parameters indicating its suitability for use in food processing [35]. In our study, the highest protein content (19.2%) was characteristic of emmer

wheat, as shown in Table 8. Lower values of this parameter were noted for spelt wheat (16.6%) and durum wheat (14.5%), and the lowest was observed in common wheat (13.1%). This indicates that, in spite of the high level of cultivation technology, the hulled wheat species retained a higher protein content, compared to the contemporary industrial wheat species. Wet gluten yield was strongly correlated with total protein content (coefficient of correlation of 0.94; Table 9); hence, the highest wet gluten yields were also obtained in the case of the hulled wheat species (41.8% for emmer wheat and 39.1 for spelt wheat), compared to durum (30.3%) and common wheat (29.9%).

**Table 8.** Quality traits of analysed wheat species (mean values for the years 2015–2017).

| Wheat Species | Total Protein (%) | Wet Gluten Yield (%) | Gluten Spread (mm) | Total Ash (%) |
|---|---|---|---|---|
| common | 13.1 [D#] | 29.9 [C] | 7.3 [D] | 1.68 [C] |
| durum | 14.5 [C] | 30.3 [C] | 8.7 [C] | 1.52 [D] |
| spelt | 16.6 [B] | 39.1 [B] | 9.8 [B] | 1.78 [B] |
| emmer | 19.2 [A] | 41.8 [A] | 13.3 [A] | 2.19 [A] |

[#] Values denoted with the same letter in columns do not differ statistically significantly ($p \leq 0.05$).

**Table 9.** Coefficients of correlation (r) between analysed traits of five wheat cultivars (mean values for the years 2015–2017).

| | Grain Yield | WTK | Test Weight | Grain Vitreousness | Colour (par.b *) | TPC | Wet Gluten Yield | Gluten Spread | Total Ash |
|---|---|---|---|---|---|---|---|---|---|
| Grain yield | 1 | | | | | | | | |
| WTK | 0.26 | 1 | | | | | | | |
| Test weight | 0.36 | 0.61 | 1 | | | | | | |
| Grain vitreousness | –0.18 | –0.16 | 0.16 | 1 | | | | | |
| Colour (par. b *) | 0.04 | 0.77 | 0.87 | 0.28 | 1 | | | | |
| TPC | –0.92 | –0.38 | –0.43 | 0.32 | –0.17 | 1 | | | |
| Wet gluten yield | –0.88 | –0.55 | –0.53 | 0.21 | –0.33 | 0.94 | 1 | | |
| Gluten spread | –0.85 | –0.15 | –0.41 | 0.34 | –0.03 | 0.90 | 0.80 | 1 | |
| Total ash | –0.60 | –0.55 | –0.64 | 0.36 | –0.48 | 0.72 | 0.64 | 0.66 | 1 |

WTK—Weight of 1000 kernels; TPC—Total protein content.

Subira et al. [28] analysed old and contemporary cultivars of durum wheat and arrived at the conclusion that, with the development of agriculture and cultivation technology, the content of protein in grain decreased from 16% to 14.2–14.7%. Sieber et al. [32] obtained average total protein content in durum wheat grain at a level of 11.6%. In a study by Oak et al. [36], in an experiment located in India, total protein content in the grain of emmer wheat cultivars varied in the range of 10.8–14.1%. Konvalina et al. [37] obtained high levels of total protein content for six emmer wheat cultivars (17.7–18.9%), depending on their location in the Czech Republic, and of wet gluten yield (40.9–47.7%), although the quality of the protein–determined by means of the gluten index–was not high. In another study on emmer wheat, Konvalina et al. [15] obtained higher levels of total protein content, from 16.1 to 19.0%, compared to those for common wheat cultivars (13.6–13.9%). Piergiovanni et al. [33] compared three wheat species–emmer, spelt, and durum–and noted the highest total protein content in spelt (17.1%), followed by emmer (16.7%) and durum wheat cultivars (15.3%). Wet gluten yield

values followed a similar pattern. Suchowilska et al. [12], on the other hand, obtained a higher content of total protein for *Triticum dicoccum* (mean 19.7%), compared to *T. spelta* (mean 17.0%). This agrees with the results obtained in this study, where the total protein content in emmer wheat was the highest, and regardless of the year in the study, amounted to 19.2%, compared to 16.6% in spelt. Pagnotta et al. [22] analysed 20 genotypes of emmer wheat in a two-year study cycle at two different locations in Italy and obtained identical total protein content: 19.2%.

Gluten quality can be measured by means of the parameters of gluten spread and elasticity, and is a property which affects the properties of dough and bread, and has an impact on the quality of pasta. In our study, gluten spread in durum wheat grain was 8.7 mm (Table 8). Hulled wheat species, even though they presented higher levels of wet gluten yield, had lower quality gluten, being characterised by higher spread values. In the case of spelt wheat, the value of this parameter was noted at a level of 9.8 mm, and in the case of emmer wheat, the highest spread value (being the limit value) was obtained: 13.3 mm.

### 3.7. Total Ash Content

The highest ash content, which is related to the mineral levels in the raw material, was noted in emmer wheat grain (2.19%), followed by spelt (1.78%) and common wheat (1.68%), while being the lowest in durum wheat grain (1.52%). The results obtained correspond with the results of studies by other authors. Total ash content estimated by Rachoń and Szumiło [38] for durum wheat was 1.8–2.0%, for spelt wheat was 1.7%, and for common wheat was 1.6%. In a study by Sieber et al. [32], total ash content in durum wheat grain was in the range of 1.70–2.05%. Sobczyk et al. [39] obtained total ash content at a level of 1.58–2.40% for 10 spelt wheat cultivars, and 1.17% for common wheat. They concluded that the higher value in the case of spelt resulted from a higher share of seed coat in the spelt wheat kernel, compared to the kernel of common wheat. Pagnotta et al. [22] studied 20 genotypes of emmer wheat and obtained total ash content in the range of 2.0–2.4%, with a mean value of 2.2%; Desheva et al. [25], in the case of 39 genotypes, obtained total ash content in the range of 1.66–2.27%.

### 3.8. Correlation of Analysed Traits

Total protein content was very strongly and positively correlated with wet gluten yield ($r = 0.94$). In turn, strong and positive correlations were noted between total protein content and wet gluten yield and gluten spread ($r = 0.90$ and $r = 0.80$, respectively); between total protein content total ash content ($r = 0.72$ and r = 0.64, respectively); and between grain colour and WTK and test weight ($r = 0.77$ and $r = 0.87$, respectively). A weak correlation was noted between total protein content and gluten spread and grain vitreousness ($r = 0.32$ and $r = 0.34$, respectively). Test weight was significantly and positively correlated with the weight of 1000 kernels ($r = 0.61$), and negatively correlated with total ash content in grain ($r = −0.64$); see Table 9. Negative and very strong correlations were noted between grain yield and total protein content, wet gluten yield, and its spread ($r = −0.92$, $r = −0.88$, and $r = −0.85$, respectively). Significant negative correlations were noted also between grain yield and total ash content ($r = −0.60$), and between grain colour and gluten content and ash content ($r = −0.33$ and $r = −0.48$, respectively).

### 3.9. Stability of Traits

Analysis of the coefficients of variation (Table 7) revealed that, among the analysed traits, regardless of the wheat species, lower variation was characteristic of test weight (from 1.09% for common wheat to 2.60% for emmer), total protein content in grain (from 2.46% for emmer to 5.14% for spelt), wet gluten yield (from 2.98% for emmer to 9.03% for common wheat), weight of 1000 kernels (from 2.44% for emmer to 8.47% for spelt), and grain yield (from 4.81% for durum to 6.18% for spelt).

Higher variation was noted in the case of such traits as gluten spread (from 20.27% for common wheat to 8.39% for emmer), grain vitreousness (from 26.34% for spelt to 1.67% for emmer), and total ash content (from 15.08% for emmer to 3.75% for common wheat).

Among the analysed wheat species, the lowest variation was noted in emmer wheat (lowest variation in three traits: WTK, grain vitreousness, and total protein content). The highest variation of traits was observed in the case of the spelt wheat (highest coefficients of variation for four traits: WTK, test weight, grain vitreousness, and total protein content).

## 4. Conclusions

In this study, we demonstrated that hulled wheat species, when cultivated under a high level of cultivation technology, produced distinctly lower yields as compared to the conventionally cultivated wheat species common wheat and durum wheat: yields were lower, in the case of spelt, by 42% and 30%, and in the case of emmer, by 56% and 46%, respectively. However, in spite of their lower yields, the hulled wheats yields were stable under variable agro-meteorological conditions in subsequent years. Both tested hulled wheats retained very good quality parameters, significantly surpassing the naked wheat species. In particular, high grain vitreousness, total protein content, wet gluten yield, and total ash content were characteristic of emmer wheat; however, its gluten spread was also the highest (13.3 mm), and therefore, its application as a raw material in certain branches of food processing may be limited. In summary, hulled wheat species can be recommended for contemporary agricultural production as an alternative source of high-quality raw material for the agricultural and food industries.

**Author Contributions:** Conceptualization, A.B.-M. and L.R.; methodology, A.B.-M. and A.K.-D.; software, A.B.-M.; validation, L.R.; formal analysis, A.B.-M. and A.K.-D.; investigation, L.R.; resources, A.B.-M.; data curation, A.K.-D.; writing-original draft preparation, L.R.; writing—review and editing, A.K.-D. and A.B.-M.; visualization, A.K.-D.; supervision, L.R.; project administration, A.B.-M.; funding acquisition, L.R. All authors have read and agreed to the published version of the manuscript.

**Funding:** This research received no external funding.

**Conflicts of Interest:** The authors declare no conflict of interest.

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
