# Peer review of "Hulled Wheat Productivity and Quality in Modern Agriculture Against Conventional Wheat Species"

_agriculture, doi:10.3390/agriculture10070275_

Round 1

Reviewer 1 Report

Comments/Questions

The manuscript entitled “Hulled wheat productivity and quality in modern agriculture against conventional wheat species” deals with the evaluation and comparison of the yields and quality of hulled wheat species, emmer  wheat  and  spelt  wheat,  with  commonly  grown  naked  wheat  species  –  common wheat and durum wheat. All the mentioned wheat species were subjected to a field experiment with identical cultivation conditions, and the applied cultivation technology was in conformance with the principles of cereal production in contemporary agriculture.

The English language and style are correctly used and written.

The article is of much interest to the readers of Agronomy, and should be adequately revised before accepted for publication.

Minor revision

Line 64: … and durum wheat. (Finish the sentence without comma punctuation mark.)

Line 66: … the above mentioned wheat species … (not abovementioned)

Lines 79 -82: Check if the wheat (Triticum L.) species are correctly written, and do the durum and emmer species correspond to previously written in Introduction section?

Line 116: Was the measuring of the grain colour parameter b* done on whole grain samples or the grains were milled to obtain whole-grain flour before measuring the colour? My opinion is to write this method description with more clarity.

Line 129: Check … the coefficients of correlation → variation?

Line 180: … lower yields of hilled wheat species … → hulled wheat species?

Line 208: Make distinction between letters written in lowercase and UPPERCASE, as they refer to different statistical significance, in rows and columns I presume.

Line 248: Missing word … test weight of grains.

Line 382: … wet gluten ash → wet gluten yield?

Author Response

Review 1. Response

Thanks a lot to the Reviewer for a detailed analysis of our manuscript and constructive comments. All comments were taken into account when proofreading work. We've thoroughly corrected:

Line 64: we removed the comma as suggested by the reviewer

Line 65: we have inserted a space to separate 'above' from 'mentioned'

Lines 79 -82: We spelled the durum and emmer wheat name, now are correct and correspond to previously written in Introduction section: Line 80: - T. durum Desf. and Line 82: T. dicoccum Schübl.

Line 116:  we have added detailed information that the measuring of the grain colour parameter b* were performed on whole grain samples without comminution.

Line 130: we change the "correlation" to "variation" of course.

Line 180: instead of "hilled" we put "hulled" - it was a typo

Line 208: we made distinction between lowercase and uppercase letters in the footer of Table 4. according to Reviewer's comment

Line 248: we put the missing words "test weight of grains".

Line 382: we corrected "wet gluten ash" to "wet gluten yield" - it was our mistake

In addition, we changed the comma to a period in numbers from all tables. Such a record is characteristic of the author's' language, hence this mistake.

Reviewer 2 Report

Please read my suggestions and comments on the attached manuscript.

Author Response

Review 2. Response

Thanks a lot to the Reviewer for a detailed analysis of our manuscript and comments. Especially about the language, because English is not our native. All comments were taken into account when proofreading work. We've thoroughly corrected:

Lines: 12-22: We improved the structure all of the abstract (information, grammar ideas) according to Reviewer's comments. The grammatical and structural unsuccessful of original's Abstract may have resulted from the efforts of authors to put as much information as possible in 200 words.

Line 42: we corrected our mistake and changed "diploid" to "tetraploid"

Line 43: We changed an unfortunate sentence regarding the history of spelt wheat cultivation into the following: "It was one of the earliest cultivated wheat species just after to einkorn".

Line 48: according to Reviewer's question, we added that it is about the human organism

Line 50-52: Referring to the reviewer's question: "I do hope that there is a food surplus, then why a billion people go hungry every day?" There is a World regions when is food surplus and there is a World regions when people go hungry. This is due to the poor global distribution of food and the political economy. The region that the authors represent has long been not fighting hunger, but the poor quality of large amounts of food. That is why it is so important to care for the nutritional quality of food and the environment. Yield is no longer the most important thing for us. However, we converted the questionable sentence so that our publication was global and did not comment on the problem of hunger: "Nowadays, when sustainable agriculture is the preferred most than yields, hulled wheat species can be an excellent alternative for producers of high-quality food."

Line 70: We removed the incomprehensible word "strict". In fact, it is only used in translations from the Polish language.

We have also improved subsequent language-specific translations to more global ones:

Line 83: conventional tillage system instead of "plough system"

Line 91: Protection treatments against diseases and pests ...

Line 108: "in the beginning of August" instead of "in the first decade of August"

Additionally, we changed the comma to a period in numbers in all tables. Such a record is characteristic of the author's' language too.

So far, we have not mentioned the impact of climatic conditions on the yielding of tested plants, which the reviewer rightly noted in the commentary. So we added in: Lines 179: However, for hulled wheats a yield was stable regardless of weather conditions. Both the emmer and spelt yields did not differ statistically in particular years of the study. While the yielding of high-yielding wheat - common and durum - has changed significantly depending on the agrometeorological conditions.

In addition, we have also extended Conclusions to include this statement: However, in spite of the lower yields the hulled wheats yields were stable, regardless of variable agro-meteorological conditions in their subsequent years.

Round 2

Reviewer 2 Report

Although manuscript was slightly improved, however, it is of medium quality in terms of design, results and discussion.

Author Response

Dear Reviewer

Thank you very much for your feedback. If you expected the manuscript to be corrected, you should have given directions. It's hard to guess what you mean. We hope that after reviewing the manuscript again you will be willing to put "Yes". In our opinion, it does not differ from the level of others published in Agriculture.